# Factors associated with U.S. adults' willingness to allow teenagers to play tackle football

**Kyle A. Kercher** [ID]¹*, **Jonathan T. Macy**², **Dong-Chul Seo** [ID]², **Jesse A. Steinfeldt**³

**1** Department of Kinesiology, School of Public Health-Bloomington, Indiana University, Bloomington, IN, United States of America, **2** Department of Applied Health Science, School of Public Health-Bloomington, Indiana University, Bloomington, IN, United States of America, **3** Department of Counseling and Educational Psychology, School of Education, Indiana University, Bloomington, IN, United States of America

* kylkerch@iu.edu

## Abstract

Little is known about the individual factors, such as knowledge and attitudes (*i.e.*, football safety knowledge, football attitudes), related to adults' willingness to allow adolescents to participate in tackle football. To address this gap, this study examined the extent to which football safety knowledge and attitudes toward head injury risk are associated with adults' willingness to allow teenage boys to play high school tackle football. Data were obtained from an internet-based survey of a nationally representative sample of U.S. adults aged 18 to 93 years (n = 1,018). We conducted multilevel linear regression modelling to examine independent effects of the football safety knowledge- and attitude-based predictors. Our analyses revealed that knowledge of football safety measures, along with four of the five attitude-based variables were significantly associated with adults' willingness to allow teenagers to participate in tackle football, over and above demographic factors. This study provides the first nationally representative examination of willingness to allow tackle football participation while extending our understanding of the gap between policy, public perception, and behavior present in U.S. high school football. These results point to promising directions for stakeholders aiming to increase tackle football participation as an increased understanding of the factors associated with participation may help inform effective policy-making, intervention design, and parental decision making.

## Introduction

In 2019 the Centers for Disease Control and Prevention reported that sport participation remains low with 46% of youth not participating on any sports team in the past year [1]. A range of national initiatives from organizations such as the U.S. Department of Health and Human Services [2], Aspen Institute [3], and U.S. Olympic Committee [4] have been implemented in the past decade to help take advantage of the public health opportunity provided by youth sport. Despite the promise of these initiatives, segments of the population are disproportionately affected [5]. For example, youth from low-income households are up to 23.5% less likely to have engaged in any sport activity during the past year compared to children from households with annual income greater than $100,000 [5]. In the context of youth football,

**Data Availability Statement:** All relevant data are within the manuscript and the dataset has been added to Supporting Information files.

**Funding:** The author(s) received no specific funding for this work.

**Competing interests:** The authors have declared that no competing interests exist.

USA Football, the national governing body over amateur football in the U.S., started the Football Development Model aiming to make the game safer through reduced contact and age-appropriate coaching [6]. Albeit relatively new, these initiatives signal preliminary momentum for participation in youth sports, such as tackle football to develop as a national public health priority.

Although driven primarily by male participation, football remains the second most popular high school sport among boys and girls in the U.S., and it is the most popular boys' sport by approximately 400,000 participants [7]. Part of this popularity stems from distinctive features of football in that it is accessible to both rural and urban populations [8], inclusive of a wide range of body types [9], and incorporates large team sizes. For these reasons, along with the thoroughly engrained nature of football in American culture, high school tackle football provides a fertile ground for nationally representative investigation of the factors associated with participatory decisions that may have scalable implications for youth public health outcomes [10].

While there are positive aspects associated with tackle football, the landscape in the U.S. is shifting. Nine of the past ten years have seen declines in high school tackle football participation [7]. These declines may be related to increased awareness of the relatively high head impact frequencies and magnitudes in tackle football along with initial work suggesting that earlier participation in tackle football may be associated with worse later-in-life cognitive and behavioral outcomes [11, 12]. These significant strides in brain injury research and increased societal awareness have coincided with regulative changes related to tackling, equipment innovations, practice structural recommendations, and game rules at various levels [11–14]. Notably, there are six states with active or pending legislation limiting the amount of allowable weekly full-contact practice time and/or banning tackle football before a certain age [15]. These moves are historic because aside from concussion legislation, the youth football environment has typically been regulated by a complex sport governance system with minimal involvement by state legislatures [16]. However, while these regulative changes targeting increased safety and national initiatives emphasizing participation are promising for the future of football participation, little is known about the factors driving football participatory decisions at the individual level (*e.g.*, knowledge, attitudes).

To our knowledge, there has not been a nationally representative assessment of U.S. adults' willingness to allow tackle football participation; nor has there been an assessment of the relationship between individual-level behavioral constructs (*e.g.*, knowledge, attitudes) and willingness to allow tackle football participation. Previous research has explored individuals' willingness to allow children to play football, but those studies used convenience samples of college students and respondents from only one state [17, 18]. This is a significant limitation in the current literature because youth sport participatory outcomes have been shown to differ significantly across demographic groups [5]. Chrisman et al. [19] explored support for tackling restrictions in youth football, a related topic, but supporting tackling restrictions is not as bold or influential of a parenting decision as willingness to allow teenage boys to play tackle football. Past research has also argued for greater empirical attention on adults in the youth sport context (*i.e.*, coaches, parents, etc.) as they have been shown to play a key role in determining the youth motivational climate and sustained youth sport participation [20–22]. Consequently, the present study was warranted in that it addressed both the underrepresented topic of youth football participatory decisions in adults and incorporated a nationally representative sample. The research question guiding the present study was: After controlling for demographic influences, to what extent are knowledge of football safety measures and attitudes toward head injury risk associated with adults' willingness to allow teenage boys to play high school tackle football? Our two primary hypotheses were that (1) respondents with greater knowledge of

football safety measures would be more willing to allow teenage boys to participate in high school tackle football and (2) respondents with greater worry about head injury risk (i.e., attitude) would be less likely to allow teenage boys to participate in high school tackle football.

## Methods

### Sample

This was a prospective, cross-sectional survey study. Data were obtained from an internet-based survey of a nationally representative sample of U.S. adults aged 18 to 93 years ($n$ = 1,018). Respondents were members of Ipsos KnowledgePanel, a large probability-based online panel in the U.S., with approximately 60,000 members. The KnowledgePanel sample was developed on a foundation of address-based sampling with a statistically valid representation of the U.S. population as well as many under-researched and often harder-to-reach populations [23]. Ipsos [23] addresses self-selection bias by choosing respondents through the Delivery Sequence File of the United States Postal Service, which is a single sampling frame covering everybody in the U.S. with a postal address. Importantly, a random sample of households are sent a mail invitation to join the panel; people cannot simply volunteer. The response rate was not available but KnowledgePanel takes steps to address non-response bias (see additional information on the KnowledgePanel sampling methods by accessing their methodology report [23]). Before joining KnowledgePanel, participants were provided with written informed consent followed by a standard email invitation to the survey. This study was deemed to be exempt from full Institutional Review Board approval from the authors' university because the research posed no more than minimal risk to participants.

### Survey outcomes

Respondents were guided through demographic questions (See S1 Appendix for list of demographic items) followed by sport-specific football safety knowledge and football attitudinal items (See S2 Appendix) and sets of questions on other topics outside of the scope of the present study. The outcome variable for this study was willingness of U.S. adults to allow teenage boys play tackle football. To assess this outcome, participants responded to the following statement: *"As a parent of a teenage boy, or if I was a parent of a teenage boy, I would let him play tackle football if he wanted to."* Participants responded using a 5-point Likert scale ranging from *"Strongly Disagree"* to *"Strongly Agree."* A 5-point Likert scale was selected to provide greater measurement variance within respondents compared to binary yes/no responses. The sample was normally distributed across the outcome variable with a mean score of 3.01 (SD = 1.15; Range: 1–5), median of 3, skewness of -0.34, and kurtosis of -0.76.

There were six primary independent variables of interest used for this project that were based on knowledge and attitudes. These predictors included one knowledge of football safety measures composite variable and five attitudinal variables (*i.e.*, worry about head injuries, belief the media exaggerates the problem of concussions, belief rule changes in recent years have made football safer, belief concussions are a serious problem, and perceived prevalence of concussion).

1. ***Knowledge of football safety measures*:** We calculated a cumulative score from ten true/false questions related to various football safety measures to assess knowledge of football safety measures. These items covered measures such as tackling guidelines, equipment innovations, practice structural recommendations, and game rules at various levels. Respondents were coded with a 0 or 1 depending on whether they answered the question correctly or not, and a cumulative score out of 10 was calculated.

2. ***Worry about head injuries*:** This composite score was calculated by producing a mean score from three worry-related items. Each of the three initial questions had a question stem of *"Indicate how much you worry about each statement for your own teenage boy (or how much you would worry if you had a teenage boy)"* and covered worry about (a) concussion, (b) long-term brain health, and (c) frequency of head injuries from playing tackle football in high school (see S3 Appendix for specific survey items that were used in the analysis). The scale for each of these worry-based questions was a 4-point Likert scale assessing the extent of worry and ranged from *"I don't worry at all"* to *"I worry a lot."* These questions were a separate set of items from the knowledge and belief-based questions.

3. ***Belief the media exaggerates the problem of concussions*:** To assess this variable, respondents were provided with the following statement: *"The media exaggerated the problem of concussions in football."* A 5-point Likert scale with response options ranging from *"Strongly Disagree"* to *"Strongly Agree"* was used.

4. ***Belief rule changes in recent years have made football safer*:** To assess this variable, respondents were provided with the following statement: *"The rule changes implemented for football games in recent years have made football a safer sport."* A 5-point Likert scale with response options ranging from *"Strongly Disagree"* to *"Strongly Agree"* was provided.

5. ***Belief concussions are a serious problem*:** To assess this variable, respondents were provided with the following statement: *"Concussions in football are a serious problem."* A 5-point Likert scale with response options ranging from *"Strongly Disagree"* to *"Strongly Agree"* was used.

6. ***Perceived prevalence of concussion*:** To operationalize the attitude construct from an additional lens, we utilized respondents' perceived prevalence of concussion in high school football. This item was used in past research exploring parents' support for tackling restrictions in youth football [19]. We assessed this variable with a survey item utilizing a 0–100 scale asking respondents: *"How many players out of 100 do you estimate would get a concussion during 1 season of high school football?"*

As effectiveness and awareness of the vast number of football safety measures remains largely unexplored [14], we approached this construct in an exploratory manner. The items for football safety knowledge were developed by a panel of experts in public health policy, neuroscience, football coaching, and sport psychology (see S2 Appendix for all sport-based survey items). Additionally, the true/false structure of this assessment was similar to past literature assessing the knowledge construct within the KAB model [24–26]. Regarding the worry-based items, three items were used to create the composite score representing worry about head injuries. These items had a Cronbach's alpha = 0.96 which indicated adequate internal reliability.

Demographic items included age, gender, education, household income, and parent status (see S1 Appendix for raw demographic survey items). Gender and education utilized standard demographic categorization established by Ipsos KnowledgePanel. Gender was dichotomized as male or female. Education was recategorized into those who had attended college (*i.e.*, with or without obtaining a degree) versus those who had not attended college. Household income was recategorized into households earning less than versus more than $75,000. Parent status was utilized as a dichotomous variable based on whether respondents had a child of any age.

## Knowledge-Attitude-Behavior (KAB) model

To guide the current study, we utilized the KAB health behavior model that has been commonly employed in numerous health promotion settings to understand how social constructs

relate to health behaviors [27–30]. Some of these contexts include, but are not limited to, assessments of concussion management behaviors, doping prevention, cardiac patients, and perceptions of elder abuse [30–33]. Despite the limited research into underlying psychosocial constructs present in youth football participatory decisions, past research guided by the KAB model supported the importance of targeting both the knowledge of, and attitudes toward behavior when aiming to develop effective strategies for improving desired health-related behavior [29, 30, 32].

## Data analysis procedure

The present study conceptualized respondents as nested within their state of residence or Level-2 variable in our multilevel analysis because states provide stability and meaning to the structure of much of U.S. society through their established regulative, normative, and cultural-cognitive elements. Not only are states the nesting variable of interest in national high school sport participation data [7], but they are also the governance structure that guides competition (e.g., state associations, state legislatures). Inclusion of this nesting variable allowed us to account for the variability of willingness to allow football participation that was due to the respondents from the same geographical area being more similar to each other than respondents from different geographical areas [34].

For descriptive statistics, frequencies and percentages were computed for each categorical variable, and means and standard deviations were calculated for continuous variables. A general population weight was provided by Ipsos and applied to the data to account for different sample selection probabilities. Listwise deletion was used for handling missing data as any incomplete observations were dropped from the analysis (n = 80), which reduced the sample size to 938. Multilevel linear regression models were used to examine independent effects of the predictors on willingness to allow teenagers' participation in tackle football. We accounted for clustering of respondents within states to avoid inappropriate estimates of standard errors for the model parameters and thus misleading significance tests [34, 35]. Beginning with the null model, a series of sequential models of increasing complexity were performed following the Hox, Moerbeek, and Van de Schoot [34] recommendation for multilevel modeling. The empty model (Model 1) examined the extent to which the respondents' state explained variation in their willingness to allow football participation. Model 2, a random intercept model, evaluated whether demographic fixed effects (*e.g.*, gender, education) and parent status significantly predicted the outcome. Finally, we assessed whether the addition of the six football safety knowledge and football attitude predictors (Model 3) significantly improved model fit compared to Model 2. Analyses were performed in R 4.0.3 [36] using the lme4 package [37], full maximum likelihood was used for the estimation of these models, and the level of statistical significance was set to alpha = 0.05.

## Results

### Descriptive statistics

Table 1 shows the descriptive statistics for the nationally representative weighted sample of U. S. adults (*n* = 938) with respondents included from all 50 states and the District of Columbia. Respondents were evenly split between male and female (51.3% female), and over half of the sample attended college (62.5%).

### Model results

The state variance estimate in the null model was 0.003; the state residual variance estimate in the null model was 1.32. Thus, the intraclass correlation (ICC) was 0.0023 which means that

**Table 1. Weighted sample characteristics (*n* = 938).**

| Variable | No. (%) or Mean ± SD |
|---|---|
| Gender | |
| Male | 466 (50) |
| Female | 472 (50) |
| Age | 47.8 ± 17.8 |
| Education | |
| ≤ High School | 320 (34) |
| ≥ College | 618 (66) |
| Household Income | |
| $0 to $74,999 | 412 (44) |
| $75,000 + | 526 (56) |
| Parent | |
| No | 568 (61) |
| Yes | 370 (39) |
| Knowledge of football safety measures | 3.07 ± 2.4 |
| Worry about head injuries | 2.93 ± 1.0 |
| Belief media exaggerates the concussion problem in football | 2.25 ± 1.0 |
| Belief rule changes in recent years have made football safer | 3.21 ± 0.8 |
| Belief concussions in football are a serious problem | 4.24 ± 0.9 |
| Estimate of how many players out of 100 would get a concussion during 1 season of high school football | 25.59 ± 25.1 |
| Willingness to allow teenage boys to play football | 3.01 ± 1.2 |

0.23% of variance in respondents' willingness to allow teenagers' football participation was attributed to between-state differences. Despite the ICC being low, we proceeded with multi-level modeling because of the increased likelihood of overstating findings (*i.e.*, increased Type-I errors) due to failing to account for clustering effects of non-independent data [38]. Additionally, best practice today is to account for the clustering effect [39]. Single-level regression analyses were also conducted, and the results did not differ in any significant way.

Table 2 presents the results of the multilevel linear regression analysis of U.S. adults' willingness to allow teenagers to participate in tackle football. The multi-level model including the demographic and control variables (i.e., Model 2) produced a significantly better fit than the null model based on the Chi-square test of deviance ($\chi^2(5) = 27.63$; p < .001). Additionally, the multilevel model with the six football safety knowledge and football attitude variables (i.e., Model 3) produced a significantly better fit to the data than Model 2 ($\chi^2(7) = 358.07$; p < .001). The $R^2$ for Model 2 (i.e., demographic model) was 0.029, whereas the $R^2$ for the final model (Model 3) was 0.340. Lower Akaike information criterion (AIC) and Bayesian information criterion (BIC) values from Model 3 (AIC: 2612.2; BIC: 2680) compared to Model 1 (AIC: 2975.9; BIC: 2990.4) and Model 2 (AIC: 2958.3; BIC: 2997.0) also indicated that Model 3 was the best fitting model. Notably, since the difference in AIC and BIC is greater than 10, there is "very strong" evidence for the more complex Model 3 [40, 41].

No significant demographic contributions were maintained in the final model (Model 3). Although females (Est = -.36; SE = .07; p = < .001) were less willing to allow teenage boys to play high school tackle football compared to males in the demographic model (Model 2), this influence was no longer statistically significant once knowledge and attitude variables were added. Notably, being a parent (Est = .08; SE = .07; p = .212) was not associated with willingness to allow teenage boys to play tackle football.

**Table 2. Willingness to allow high school tackle football participation regression models.**

| | Model 1 | | Model 2 | | Model 3 | |
|---|---|---|---|---|---|---|
| **Variable** | **β Est (SE)** | **p value** | **β Est (SE)** | **p value** | **β Est (SE)** | **p value** |
| *Fixed effects* | | | | | | |
| Intercept (constant) | 3.02 (0.04) | <0.001 | 3.35 (0.15) | < .001* | 1.62 (0.26) | < .001* |
| Gender | | | | | | |
| Male | | | Reference | | Reference | |
| Female | | | -.36 (0.07) | < .001* | -0.03 (0.07) | .613 |
| Age | | | -0.002 (0.002) | .309 | -.0003 (0.002) | .902 |
| Education | | | | | | |
| ≤ Graduated high school | | | Reference | - | Reference | - |
| ≥ Attended college | | | -0.11 (0.08) | .188 | -0.07 (0.07) | .279 |
| Household income | | | | | | |
| $0 to $74,999 | | | Reference | - | Reference | - |
| $75,000 + | | | 0.06 (0.08) | .465 | -0.002 (0.07) | .977 |
| Parent | | | | | | |
| No | | | Reference | - | Reference | - |
| Yes | | | 0.008 (0.08) | .923 | 0.08 (0.07) | .212 |
| Knowledge of safety measures | | | | | 0.04 (0.01) | .002* |
| Worry about head injuries | | | | | -0.27 (0.03) | < .001* |
| Belief media exaggerates the concussion problem | | | | | 0.38 (0.03) | < .001* |
| Belief recent rule changes have made football safer | | | | | 0.37 (0.04) | < .001* |
| Belief concussions in football are a serious problem | | | | | 0.02 (0.04) | .607 |
| Estimate of how many players out of 100 would get a concussion during 1 season of high school football | | | | | -0.002 (0.001) | .053 |
| *Random effect* | | | | | | |
| State residual variance (SD) | 1.32 (1.15) | | 1.28 (1.13) | | 0.87 (0.93) | |
| Model testing: $\chi^2$ (df) | | | 27.63 (5) | <0.001* | 358.07 (6) | <0.001* |

*Note*: Outcome variable: willingness to allow high school football participation; β Est = standardized beta coefficients; SE = standard error. Each model was compared to the immediate previous model.

*p < .05.

Four of our primary predictors of interest achieved significance and aligned with our a priori hypotheses. Knowledge of football safety measures (Est = .04; SE = .01; p = .002), worry about head injuries (Est = -.27; SE = .03; p < .001), belief the media exaggerates the concussion problem (Est = .38; SE = .03; p < .001), and belief that rule changes in recent years have made football safer (Est = .37; SE = .04; p < .001) were all associated with the outcome. Perceived prevalence of concussion (Est = -.002; SE = .001; p = .053) was not a significant predictor. Contrary to our hypotheses, belief that concussions are a serious problem (Est = .02; SE = .04; p = .554) was not statistically significant.

A sensitivity analysis was conducted by running the same final model on a sample that included only parents and the pattern of results was highly similar. The only difference in any collected variables was that education (Est = -.25; SE = .11; p = .019) became a significant predictor.

## Discussion

While substantial national initiatives have begun to make youth sports a public health priority, little is known about the association between adults' football safety knowledge and football attitudes and willingness to allow adolescent boys to participate in tackle football. Few studies have examined this gap in tackle football and those that have were limited by utilization of convenience samples [17, 18] or the outcome of support for age-based tackling restrictions [19], which is a rule change rather than the more generalizable public health decision of willingness to allow participation. To address this gap in the literature, along with calls for greater empirical attention to be placed on the role of adults in youth sport participatory research [21, 22], we examined the association between adults' football safety knowledge and football attitudes about issues specific to tackle football and willingness to allow teenage boys to play tackle football, over and above the contribution of demographic factors.

There were three key findings from this study. First, worry about head injury risk and perceived prevalence of concussion were negatively associated with adults' willingness to allow tackle football participation. Second, belief that rule changes in recent years have made football safer and that the media has exaggerated the problem of concussions in football were both positively associated with the outcome of interest. Third, adults' knowledge of football safety measures was associated with support for teenage boys playing high school tackle football. This study provides the first nationally representative examination of willingness to allow tackle football participation while extending our understanding of the gap between football safety policy, public perception, and behavior present in U.S. high school football.

In line with the KAB model which proposes that attitudes play an important role in projecting behavior [27, 29, 30], the current study found support for our attitude-based hypothesis that respondents with greater worry about head injury risk would be less likely to allow teenage boys to participate in high school tackle football. Within this hypothesis, three of our five attitude-based variables significantly predicted the outcome of interest, and a fourth attitude-based predictor had a p-value of 0.053. Among these four attitude-based predictors, the two that were negatively associated with willingness were worry about head injuries and perceived concussion prevalence. Adult worry about head injuries is particularly important because the parent-initiated motivational climate plays a critical role in determining youth athletes' motivation [20–22]. Our finding of a link between adults' attitudes about head injury risk and willingness to allow football participation provides additional support for the importance of this parent-initiated motivational climate. If adults' hold more negative attitudes toward head injury risk, then they were less likely to allow participation. Therefore, in hopes of increasing engagement and adherence in football, we may want to place a significant emphasis on addressing the attitudes of prominent adults (e.g., parents, coaches, teachers) toward the actual, rather than perceived, risks and rewards of football participation.

Perceived concussion prevalence was another predictor of decreased willingness to allow tackle football participation. This estimate is noteworthy because it points to the importance of education-based interventions that aim to inform the public on accurate risk involved in football participation. The mean score of perceived concussion prevalence in our study was 25.6 per 100 athletes, which is a largely overestimated assessment of concussion risk in a representative sample of U.S. adults. This is concerning because past research suggested the expected prevalence was approximately 5 per 100 athletes [42] and more current tackle football research published in 2021 supported a rate of 6.99 concussions per 10,000 athletic exposures [43], both of which are well below the 25.6 per 100 athletes estimate seen in this study. It is possible that the overinflated concussion estimate is related to the increased media attention concussions have garnered over the past 15 years [18, 19], which is likely to influence attitudes regarding

football participation. Future youth football campaigns may aim to include an educational component targeting the spread of accurate concussion risk information to allow parents to make decisions based on the most accurate data possible.

Our second key finding was that both belief that rule changes in recent years have made football safer and that the media has exaggerated the problem of concussions in football were significantly associated with increased willingness to allow tackle football participation. As there have been a myriad of modifications to tackle football at all levels over the past decade, it is promising for the future of football to see that there is an association between perception of rule changes and willingness to allow participation. While comprehensive evaluation of most rule changes remains scarce [14], identification of this association bolsters the idea of educating the public on rule changes and corresponding injury risk reductions as a strategic option for increasing football participation. The other positively associated attitude predictor was belief that the media has exaggerated the problem of concussions in football. This was in line with our hypothesis and further strengthens the premise of targeting various types of adult attitudes as a strategy to increase youth football participation. Unfortunately, since fear, tragedy, and misinformation spread much faster and more widely than accurate news [44, 45], disseminating information about football safety improvements in the media is likely to continue to be challenging. Taken together, these two significant findings about rule changes and media exaggeration point to the importance of football stakeholders continuing to emphasize evaluation and improvement in their communication strategies to the public. We do not envision the challenges of navigating the media and disseminating accurate safety information going away any time soon as game rule and practice structure guideline changes are likely to become even more important as researchers continue to learn more about the implications of subconcussive head impacts on individuals' long-term brain health [11, 46–48].

The third key finding from the present study was that greater knowledge of football safety measures was associated with increased willingness to support tackle football participation. This is consistent with the KAB model and supports our hypothesis that respondents who are more familiar with football safety policies are more willing to support tackle football participation. While this finding is significant, the low mean score of 3.07 out of 10 also indicates that the average adult may not have good knowledge of football safety measures. Related to the low knowledge score, past literature found a gap between policy mechanisms (i.e., safety measures) and behavioral outcomes [49] which may be present in the football context as well. In other words, just because safety policies, such as constantly evolving concussion legislation or game rule changes, are enacted that does not mean that stakeholders are aware of them or that they will succeed with their intended impact on behavioral outcomes. Tackle football provides a fitting example of this complex problem as the sport has seen a tremendous amount of evolution in safety policy over the past two decades [14]. However, according to Macy [14], their efficacy often remains unclear and the relatively low knowledge scores may point to a need for increased education of stakeholders (*i.e.*, parents). Policymakers may be well served to invest in efforts designed to educate their target populations of their proposed policies. If they can educate more people on their policies, then our results suggest more people may be willing to allow participation in football, which is one of the primary intents behind the football safety policies.

While our primary aims were related to psychological predictors, the demographic findings warrant brief mention as one finding deviated from past research that found that females were more likely to support tackling restrictions [19]. In our sample, none of the demographic factors were significant in the final model when psychosocial predictors were added to the model. Future studies may aim to examine these demographic predictors more thoroughly, but the explanatory mechanisms are beyond the scope of this study.

While there were substantial findings with practical implications derived from this study, there were also significant limitations. First, we cannot infer causation since this was a cross-sectional assessment. Second, the sample was comprised of adults rather than only parents. It is possible that the study findings would have been different if the national sample was made up of only parents who may have been able to think more realistically about decisions for their children rather than their theoretical children. However, our sensitivity analysis with a sub-sample of parents produced the same pattern of findings as the full sample. Third, the football safety knowledge quiz was utilized as a proxy measure, may have been too difficult as an assessment tool for comprehensive knowledge of football safety measures. Fourth, while the results from this study point to football safety knowledge and attitudes toward injury risk being important factors related to the outcome variable, there are likely other critical factors, such as normative beliefs, at play. Fifth, a 1-point Likert scale difference in willingness to allow participation is not as interpretable as the difference between yes and no, but it does show greater variance in opinions. Lastly, since the survey item asked specifically about boys, rather than all children, the results are not generalizable to all genders of tackle football participants. Despite this limitation, we anticipate these findings pointing to future research directions targeting adults' involvement in youth football participatory outcomes for participants of all genders. Future youth football research may be well served to utilize additional health behavior theories that examine additional constructs.

The results point to the importance of educating parents with accurate safety information about the risks associated with high school football participation. In a broader sense, stakeholders may want to target adults' knowledge and attitudes in their efforts to improve participatory behaviors. Doing so may help unlock all the powerful individual benefits of sport participation [3, 50–54], while creating potentially large-scale public health payoffs that can be supported by public health policy in youth football [10].

## Supporting information

**S1 Appendix. Demographic response options.**
(DOC)

**S2 Appendix. All sport based survey items (Part A).**
(DOCX)

**S3 Appendix. Survey items used in the analysis.**
(DOCX)

**S1 Table. Likert item distribution.**
(DOCX)

**S1 File. Raw data file.**
(CSV)

## Author Contributions

**Conceptualization:** Kyle A. Kercher, Jonathan T. Macy, Dong-Chul Seo.

**Data curation:** Kyle A. Kercher, Jonathan T. Macy.

**Formal analysis:** Kyle A. Kercher, Dong-Chul Seo.

**Funding acquisition:** Kyle A. Kercher, Jonathan T. Macy.

**Investigation:** Kyle A. Kercher, Jonathan T. Macy, Dong-Chul Seo.

**Methodology:** Kyle A. Kercher, Jonathan T. Macy, Dong-Chul Seo, Jesse A. Steinfeldt.

**Project administration:** Kyle A. Kercher.

**Resources:** Jonathan T. Macy.

**Software:** Kyle A. Kercher.

**Supervision:** Jonathan T. Macy, Dong-Chul Seo, Jesse A. Steinfeldt.

**Validation:** Kyle A. Kercher, Dong-Chul Seo.

**Visualization:** Kyle A. Kercher.

**Writing – original draft:** Kyle A. Kercher.

**Writing – review & editing:** Kyle A. Kercher, Jonathan T. Macy, Dong-Chul Seo, Jesse A. Steinfeldt.

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
