## [Decision Letter · Decision Letter 0]

11 May 2022

PONE-D-21-23107Factors associated with U.S. adults' willingness to allow teenagers to play tackle footballPLOS ONE

Dear Dr. Kercher,

Thank you for submitting your manuscript to PLOS ONE. After careful consideration, we feel that it has merit but does not fully meet PLOS ONE’s publication criteria as it currently stands. Therefore, we invite you to submit a revised version of the manuscript that addresses the points raised during the review process.

The reviewer's have addressed several key points that should be addressed in your revised manuscript. More specifically, please review each section to ensure the appropriate placement of the content (introduction, methods, results, and discussion).  Also, more detailed description is needed for the chosen methodology. Please see Reviewer 1's comments. 

We look forward to receiving your revised manuscript.

Kind regards,

Jacob Resch, Ph.D.

Academic Editor

PLOS ONE

Journal Requirements:

2. Please provide additional details regarding participant consent. In the Methods section, please ensure that you have specified (1) whether consent was informed and (2) what type you obtained (for instance, written or verbal). If your study included minors, state whether you obtained consent from parents or guardians. If the need for consent was waived by the ethics committee, please include this information.

3. PLOS requires an ORCID iD for the corresponding author in Editorial Manager on papers submitted after December 6th, 2016. Please ensure that you have an ORCID iD and that it is validated in Editorial Manager. To do this, go to ‘Update my Information’ (in the upper left-hand corner of the main menu), and click on the Fetch/Validate link next to the ORCID field. This will take you to the ORCID site and allow you to create a new iD or authenticate a pre-existing iD in Editorial Manager. Please see the following video for instructions on linking an ORCID iD to your Editorial Manager account: https://www.youtube.com/watch?v=_xcclfuvtxQ.

Reviewers' comments:

Reviewer's Responses to Questions

**Comments to the Author**

1. Is the manuscript technically sound, and do the data support the conclusions?

Reviewer #1: Yes

Reviewer #2: Yes

2. Has the statistical analysis been performed appropriately and rigorously? 

Reviewer #1: Yes

Reviewer #2: Yes

3. Have the authors made all data underlying the findings in their manuscript fully available?

Reviewer #1: Yes

Reviewer #2: Yes

4. Is the manuscript presented in an intelligible fashion and written in standard English?

Reviewer #1: Yes

Reviewer #2: Yes

5. Review Comments to the Author

Reviewer #1: Lines 18-21: Consider breaking this into two sentences, with the 2nd sentence starting as "Little is known...". This will be more direct for readers. Line 20: Please specify knowledge and attitudes here (i.e., football safety knowledge and football attitudes) for clarity.

Line 24: Normally I do not make formatting edits, but believe PLOS One does publish online ahead of print. So, I am providing them where they stick out to provide a cleaner copy for readers. Specifically here, please remove the line break and paragraph indentation for the abstract.

Line 26: Similar comment to above, please consider referring to "knowledge" and "attitudes" here and throughout as "football safety knowledge" or similar for reader understanding. I believe this will help comprehension throughout.

INTRODUCTION: Overall this section provides sound background, rationale, and importance of this study. However, the flow and paragraph use is somewhat confusing, and may be a matter of field/discipline preferences. Edits are provided below to help provide funneling flow.

Lines 40-41: Consider revising to "...CDC reported that only 26% of youth met this recommendation."

Line 42: Consider specifying the year(s) the 46% statistic comes from. Mentioning because we are in a COVID-era and physical activity, specifically sport, has been impacted.

Lines 47-49: Similar comment; please specify year so it's clear to readers.

Line 59: Typo in this sentence at this point; believe changing "have developed" to "have been implemented..." will work.

Lines 78-83: Consider inserting a couple sentences specifically addressing the concerns of repetitive head impacts/concussion occurring in youth sports. For example, there was initial work suggesting football participation before 12 years old was associated with worse late-life cognitive and behavioral outcomes (Stamm et al. 2015; Neurology). Further, there are relatively high head impact magnitudes (e.g. linear and rotational acceleration) and head impact rates (e.g. number of head impacts players experience in a typical practice or competition) experienced in youth tackle football. Lastly, speaking to the national legislation rather than 1 state may provide more implications. For example, there are currently 6 states with active/pending legislation for minimizing contact practice times and/or banning tackle football before a certain age (see https://brainlaw. com/youth-tackle-football/). I believe this will enhance the rationale for "why" this study is needed.

Lines 86-89: Consider adjusting the flow of this introduction so that your research question comes in the final paragraph of the introduction. As is, the research question is provided, and then further rationale behind it is provided. It may be easier for reviewers to be funneled from the overarching context, the underlying rationale, and then finally to the specific research question as is common in many fields.

Lines105-122: This information is a bit long and unnecessary to the majority of readers, and also may be more appropriate in the methods section (since this is justification for a methodological decision). The authors should consider summarizing it into ~2 sentences stating (paraphrasing) that they used a KAB model which has been commonly employed in numerous health promotion applications to understand how unique social constructs relate to health behaviors. As a reader and reviewer, this is much more direct and gets the same point across. In the limitations, you can also acknowledge this decision was used and the mixed evidence surrounding its general applicability.

Lines 123-127: Please remove this information as nothing new is added with this text that is not already mentioned in the intro/methods already.

Lines 127-130: Consider moving these hypotheses right after your specific aims/research question for improved flow.

METHODS: The methods overall are well-written. The reordering of paragraphs and some further details are warranted. Lastly, greater details and justification for the statistical modeling are warranted due to points raised below.

Line 133: Please specify this was a "prospective, cross-sectional survey study." This will be useful information for readers, as well as future research systematically reviewing work on this topic.

Lines 136-137: Please 1) move this text to the data analysis section to better inform readers this was performed, and 2) make sure the language surrounding population weighting matches the text on lines 236-238.

Lines 156-157: Please also add the median score here for further support of the Gaussian distribution.

Lines 150: Please consider: 1) changing the "measures" here to "Survey Outcomes". 2) provide any supporting evidence for the development, reliability, validity of this survey for addressing this research question. As is, it is (I believe) touched upon within each predictor's paragraph by specifying a Cronbach's alpha and other text, but believe a standalone paragraph with all this information (likely at the end of the survey outcome section) would be more appropriate. 3) state the order in which all the survey items/sections were administrated so it could be replicated in the future. It would be beneficial to have a couple lines of text explicitly stating the order and components of the survey as the "Survey Outcomes" opener, and then provide the survey item paragraphs in the order they were presented to respondents.

Lines 154-155: Can the author please provide their rationale for using a 5-point Likert scale here rather than a binary yes/no for this outcome variable? I foresee this being a point of criticism for interpreting the findings and their implications For example, what do the beta-values presented in Table 2 associating with a 1-point Likert change in willing to allow tackle football participation mean for stakeholders? What is considered an important change? 1 or more-Likert point, any change?

Line 181: Why was the "worry about head injuries" category the only one examined on a 4-point Likert?

Line 176: As mentioned, I assume this is for reliability/agreement assessed by the authors for these items, but without specific detail it is unclear. A paragraph targeting the survey reliability/validity would be fruitful.

Line 217: the authors mention "full maximum likelihood here" but also mention R and the lme4 package. Mentioning because lme4 used restricted maximum likelihood by default, and want to confirm accurate reporting. Additionally, this sentence should be moved down towards the end of the paragraph as it is out of place when describing descriptive statistics in the next paragraph.

Lines 207-215: Please move this paragraph into the "Data Analysis Procedure" section as it is not relevant to the outcome measures.

Lines 201-206: Please confirm the dichotomization of these outcomes was through survey respones (i.e. binary response) rather than data processing by the authors. If data processing by the authors, please provide the exact response options for these so this survey could be replicated. The authors should consider modifying the S1 appendix to include the entire survey contents rather than just the predictor/outcome variables for full transparency.

Line 233: Please revise "p<0.05" to "alpha=0.05" because your threshold is a set value and the p-value threshold is set by your alpha value.

RESULTS: Findings are accurately reported in relation to the models ran. However, concerns are raised about the model appropriateness given the negligible variance attributed to the state levels. Further details are provided below.

Lines 243-250: The authors report the increased likelihood of Type I error as justification for using the multi-level model. However, the ICC value of 0.0023 is considerably smaller than those reported by the referenced simulation study, whose minimum ICC value used was 0.01 (over 4x larger). Without data supporting this decision to include such a small ICC and variance associated with the state level (0.003), it does not appear mulit-level modeling is necessary here. This may also increase your likelihood for Type II error. Please consider and revise if necessary.

Table 1. Please describe how fractions of a frequency are being derived, specifically for the demographic counts. Is this due to the national weighting applied? Please confirm this level of decimal precision is needed, otherwise, please round to nearest whole number for frequencies to avoid confusion.

Table 1. Though some of the predictors/outcomes appear to be normally distributed, it may be optimal to also report their median and range, or the frequency (%) for each Likert category for full transparency.

Table 2. Please remove the "note" section (leave abbreviations) in this table, and instead change the table title to "Willingness to Allow High School Tackle Football Participation Regression Models.

DISCUSSION: The discussion section provides accurate interpretation of the findings observed. However, the noted statistical model concerns have the potential to alter the findings. Thus only broad review is provided here as a 2nd round of review is warranted.

Lines 327-328: The reference indicating 5 concussions per 100 athletes is an outdated (published in 2000) sport epidemiology report. Sport concussion prevalence is considerably higher now, though is not presented as # per athlete as this fails to account for athlete-exposure. Please see publication by Chandran et al. 2021 in the American Journal of Sports Medicine for more current concussion rates.

Line 336: Here and throughout, the directionality of the outcome variable association should be provided (e.g. ".... were significantly associated with [increase? decrease?] levels of football participation willingness."

Lines 359-360: The authors should also consider that the survey used may have been too difficult and thus not an accurate measure of football safety knowledge. As a concussion and head impact biomechanics researcher, I would have only gotten 8/10 items correct.

Lines 398-403: Consider either removing or adding this note text to where it is referenced. Superscripted notes are often ignored or missed by readers, and thus should be placed above where applicable.

Reviewer #2: General Feedback:

Thank you for the opportunity to review this manuscript. It was very well written, easy to follow, and is an important contribution to the literature. I have suggested only minor revisions as detailed below.

Specific Feedback:

Introduction –

Very well written; clear and easy to follow. The introduction develops the important narrative necessary for background understanding of the present study.

Methods –

Would the authors please report the number of individuals that were contacted by Ipsos to complete the survey so that the reader can gauge the response rate? If that information is not available, please say so.

The authors should also provide details on the length of the survey (e.g., was it just the questions for this study or were there other items included for other studies?).

Further, would the authors please report whether there were key demographic factors that were associated with missing data on the received surveys? For example, were older individuals less likely to complete the instrument?

Why were the worry-based items graded on a 4-point Likert scale when the belief-based items were rated on 5-point Likert scales?

Would the authors please clarify if the education level of “attended college (62.5%)” refers only to those who completed a degree program, or if it includes those who may have attended some college without obtaining a post-secondary degree?

Why was household income dichotomized at $75k? Was this a median split?

Results –

For transparency’s sake, I would encourage the authors to also report the frequency of each Likert scale response to all questions, either in Table 1 or in supplemental materials. I understand this is asking for some tedious work, but feel it may be useful for yours and others’ future work that may utilize similar questionnaires. I apologize if this has already been done; I was unable to successfully open the link to the appendix in the review pdf file.

I’m assuming the beta values in Table 2 are standardized. Would you please make this clear in the table note and in the main text where “Est = “ is presented?

Discussion –

The authors note that one possible explanation for overestimated concussion risk is that it might be associated with increased media attention. Were the associations between independent variables evaluated? Specifically, does perception of greater media exaggeration associate with lower perceived concussion risk in your sample? On a related note, did the authors consider interaction effects among independent variables (e.g., parentage and worries, for example)?

Thank you for addressing the concern regarding only asking about boys’ participation in football in your note.

6. PLOS authors have the option to publish the peer review history of their article (what does this mean?). If published, this will include your full peer review and any attached files.

Reviewer #1: No

Reviewer #2: No

---

## [Author Response · Author response to Decision Letter 0]

30 May 2022

Thank you so much for taking the time to review this manuscript. We have carefully reviewed each and every one of your comments and addressed them line by line in the 'Response to Reviewer Comments' document and table. Thank you for making this manuscript better through your thoughtful comments. If you'd like any additional revisions, clarifications, or are unhappy with any of our edits/responses please let us know. Thank you again for your time!

---

## [Decision Letter · Decision Letter 1]

13 Jul 2022

PONE-D-21-23107R1Factors associated with U.S. adults' willingness to allow teenagers to play tackle footballPLOS ONE

Dear Dr. Kercher,

Thank you for submitting your manuscript to PLOS ONE. After careful consideration, we feel that it has merit but does not fully meet PLOS ONE’s publication criteria as it currently stands. Therefore, we invite you to submit a revised version of the manuscript that addresses the points raised during the review process. The reviewers as well as myself appreciate your careful consideration and revision of the originally submitted manuscript. After careful consideration of your research question, results, and discussion further revision is needed. More specifically, the malalignment between the manuscript narrative, primary research question, methodology and subsequent results and discussion should be focused solely on tackle football opposed to physical activity in general prior to consideration for publication. As your survey instrument is focused solely on tackle football, the manuscript should be adjusted to focus solely on this sport opposed to "sport" or "physical activity" in general. In sum, your important findings are based solely related to tackle football opposed to sport in general and the narrative should be revised to focus on this theme. 

We look forward to receiving your revised manuscript.

Kind regards,

Jacob Resch, Ph.D.

Academic Editor

PLOS ONE

Journal Requirements:

Reviewers' comments:

Reviewer's Responses to Questions

**Comments to the Author**

1. If the authors have adequately addressed your comments raised in a previous round of review and you feel that this manuscript is now acceptable for publication, you may indicate that here to bypass the “Comments to the Author” section, enter your conflict of interest statement in the “Confidential to Editor” section, and submit your "Accept" recommendation.

Reviewer #1: All comments have been addressed

Reviewer #2: All comments have been addressed

2. Is the manuscript technically sound, and do the data support the conclusions?

Reviewer #1: Yes

Reviewer #2: Yes

3. Has the statistical analysis been performed appropriately and rigorously? 

Reviewer #1: Yes

Reviewer #2: Yes

4. Have the authors made all data underlying the findings in their manuscript fully available?

Reviewer #1: No

Reviewer #2: Yes

5. Is the manuscript presented in an intelligible fashion and written in standard English?

Reviewer #1: Yes

Reviewer #2: Yes

6. Review Comments to the Author

Reviewer #1: Thank you for the thorough revision and addressing all initial points. This reviewer feels the manuscript is acceptable as is (pending copyediting) for publication. The only additional consideration is PLOS One asks that data be made publicly available. The authors indicated "yes", but their comment on this is that "All relevant data are within the manuscript and its Supporting Information files." However, PLOS Ones policy on this is that this does not constitute data being publicly available (i.e. they want the underlying, individual datapoints/dataset be provided). If this is not possible due to IRB or other concerns, the authors should revise their statement to avoid any confusion.

Reviewer #2: Thank you for addressing my comments and those of the other reviewer. I have no further requests for revision at this time.

7. PLOS authors have the option to publish the peer review history of their article (what does this mean?). If published, this will include your full peer review and any attached files.

Reviewer #1: No

Reviewer #2: No

---

## [Author Response · Author response to Decision Letter 1]

14 Jul 2022

Thank you for the comments. Per reviewer and editor comments, we have added the data set to the submission and made edits to align the narrative, research question, results, and discussion. Please see detailed adjustments in either the Response to Reviewer Comments document or the Revised Manuscript with Track Changes document.

---

## [Editor Report · Decision Letter 2]

5 Aug 2022

Factors associated with U.S. adults' willingness to allow teenagers to play tackle football

PONE-D-21-23107R2

Dear Dr. Kercher,

We’re pleased to inform you that your manuscript has been judged scientifically suitable for publication and will be formally accepted for publication once it meets all outstanding technical requirements.

Kind regards,

Jacob Resch, Ph.D.

Academic Editor

PLOS ONE
---

## [Editor Report · Acceptance letter]

11 Aug 2022

PONE-D-21-23107R2 

Factors associated with U.S. adults' willingness to allow teenagers to play tackle football 

Dear Dr. Kercher:

I'm pleased to inform you that your manuscript has been deemed suitable for publication in PLOS ONE. Congratulations! Your manuscript is now with our production department. 

Kind regards, 

on behalf of

Dr. Jacob Resch 

Academic Editor

PLOS ONE